# Supporting Return to Work after Breast Cancer: A Mixed Method Study

**DOI:** 10.3390/healthcare11162343

**Published:** 2023-08-19

**Authors:** Nicola Magnavita, Reparata Rosa Di Prinzio, Igor Meraglia, Maria Eugenia Vacca, Gabriele Arnesano, Marco Merella, Igor Mauro, Angela Iuliano, Daniela Andreina Terribile

**Affiliations:** 1Post-Graduate School of Occupational Health, Università Cattolica del Sacro Cuore, 00168 Rome, Italy; nicolamagnavita@gmail.com (N.M.); igor.meraglia01@icatt.it (I.M.); mariaeugenia.vacca01@icatt.it (M.E.V.); gabriele.arnesano01@icatt.it (G.A.); marco.merella01@icatt.it (M.M.); igor.mauro01@icatt.it (I.M.); angela.iuliano01@icatt.it (A.I.); daniela.terribile@unicatt.it (D.A.T.); 2Department of Woman, Child and Public Health, Fondazione Policlinico Universitario Agostino Gemelli IRCCS, 00168 Rome, Italy; 3Alta Scuola di Economia e Management dei Sistemi Sanitari (ALTEMS), Università Cattolica del Sacro Cuore, 00168 Rome, Italy

**Keywords:** welfare, sleep, anxiety, depression, fatigue, work organization, disability management, barriers, facilitators, workplace

## Abstract

Breast cancer (BC) is the most common invasive cancer in the world. Most BC survivors (BCSs) continue working while dealing with cancer-related disabilities. BCSs’ return-to-work (RTW) after cancer treatment is an important stage of their recovery and is associated with a higher survival rate. In this study, we addressed the RTW of BCSs with the intention of facilitating this process through direct action in the workplace. Thirty-two women who requested assistance from January to December 2022 were enrolled in the study. Semi-structured interviews and medical examinations were conducted by a team of three physicians. Interviews were analyzed using Thematic Analysis. Moreover, a quantitative cross-sectional study was conducted to compare the health status of BCSs with that of a control group of 160 working women, using standardized questionnaires on work ability, fatigue, sleep problems, anxiety, depression, and happiness. BCSs were also asked to rate the level of organizational justice they perceived at work prior to their illness. From the qualitative analysis emerged three facilitating/hindering themes: (1) person-related factors, (2) company-related factors, and (3) society-related factors. In the quantitative analysis, BCSs had significantly higher scores for anxiety, depression, sleep problems and fatigue, and lower levels of happiness than controls. The RTW of BCSs entails adapting working conditions and providing adequate support. The work-related analysis of each case made it possible to highlight the measures that need to be taken in the workplace to promote RTW. The treatment of cancer should be paired with advice on the best way to regain the ability to work.

## 1. Introduction

Female breast cancer (BC) is the most frequently diagnosed invasive cancer, with an estimated 2.3 million new cases a year [1]. It has a rising incidence [2,3], and is the leading cause of cancer morbidity, disability, and mortality in women, worldwide [4]. Survival trends are generally increasing. The 5-year survival rate for breast cancer is currently 89.5% in Australia and 90.2% in the USA for women diagnosed between 2010 and 2014 [5], and 82% in Europe for women diagnosed between 1999 and 2007 [6]. However, there are still significant disparities internationally, with rates as low as 66.1% in India [5].

Because the peak incidence of breast cancer occurs in the working population at mid-working age, this disease causes the largest productivity loss of all malignancies in the female population [7]. Most BCSs continue working while dealing with cancer-related disabilities.

Return to work (RTW) after cancer treatment is an important part of women’s recovery and is associated with a better survival rate [8]. Work is a fundamental part of the BCSs’ “rebirth” process [9], can contribute to a new normality, and can reduce the impact of BC consequences in the survivor’s life. A better quality of life has been demonstrated in BCSs who resume work compared with those who quit their job [10]. RTW after BC is associated with physical, functional, social, and emotional well-being [11]. Unfortunately, BC is associated with a high rate of loss of employment or early retirement. A systematic review showed that the prevalence of RTW within one year of diagnosis varies from 43% in the Netherlands to 93% in the USA [12]. In conjunction with pre-diagnostic individual traits, the clinical outcome, lifestyle choices, and occupational variables are significant aspects that must be considered in relation to the RTW process [13]. The most frequently self-reported reasons for RTW failure are health impairments such as fatigue, psychological problems, memory or attention problems, pain, or sleep disturbances resulting from BC or related therapy [14,15]. Physical problems, such as lymphedema [16], cognitive problems [17], and depressive symptoms [14] are associated with impaired RTW. On the other hand, favorable working conditions, such as tailored ergonomic measures (e.g., reducing manual work) and organizational climate (e.g., support from colleagues, part-time work, graded activity, gradual return, flexibility), have a positive influence on RTW [18]. According to a summary of qualitative studies, ‘offering work flexibility’ and ‘offering work accommodations’ are among the managerial interventions found to be most effective in fostering RTW [19]. From the point of view of employers, adopting a humanistic management style by offering flexibility and increased accommodation to BCSs could be the best strategy [20].

The current management of BC patients and BCSs does not focus on either occupational exposure or working conditions [21]. Traditionally, after diagnosis and treatment of the acute illness, hospital action consists in sending directions for the continuation of chronic treatment to the attending physician. Since many chronically ill patients need to be reinstated to a work environment, it would be very useful to produce indications for the physician in charge of health surveillance in order to facilitate RTW. Since many BCSs are of working age, we were prompted to undertake a project to facilitate their return to the workplace. Three stages can be recognized in this RTW process: the first consists of an assessment of the type of work and its organization prior to BC to identify problems that might arise during RTW; the second, which consists of a personalized assessment of the worker’s condition at the end of treatment; and the third, which results in the reasonable adjustments to the work process necessary to make a sustainable and full work recovery [22]. The treatment of BC patients should be completed with indications for their return to work. The RTW of BCSs has solicited many qualitative studies. None of these, however, have been accompanied by quantitative analyses or operational proposals to promote work accommodations during the RTW process. In this study, which to the best of our knowledge is the only one in which the occupational doctors of the hospital have taken care to improve the RTW of patients, we intended to act in each of the three levels of intervention specified above, that is, both in the assessment of the previous occupational risk, in the verification of the patient’s health conditions at the end of the treatment, and finally, in the adaptation of the working modes at the return, through the occupational doctor of the company.

The School of Occupational Medicine of the Catholic University of the Sacred Heart, in collaboration with Komen Italy and the Fondazione Policlinico Gemelli IRCCS, has developed a project to promote the RTW of women with BC based on direct contact between the University’s occupational doctors and the specialist responsible for evaluating the BCSs’ fitness for work. The women who requested assistance in the RTW process underwent a medical examination to assess their physical health. They were also given a semi-structured interview and a standardized questionnaire to identify barriers and possible factors that favor their RTW. According to the results obtained, the occupational doctors of the companies where the women were employed received a series of advice that favored the reintegration of the BCSs into the workplace.

On the basis of the results of activity conducted in the first year, the aim of this study was to describe the factors that hinder or favor RTW and to analyze the health status of BCSs compared with that of working women of the same age. Indeed, both measures were necessary to develop targeted advice on how to properly manage disability in the workplace.

## 2. Materials and Methods

### 2.1. Study Design

This project was aimed at BCS patients who wanted to be supported in the RTW process and was included in the clinical care pathway provided by the Gemelli General Hospital in Rome and by Komen Italia. To find people potentially interested in the project, leaflets were disseminated in patient waiting rooms and announcements were published in a newsletter for women with BC. Occupational examinations were performed at the hospital Center for Integrated Treatments in Oncology and were free of charge. The data reported in this article refer to observations collected in 2022—the first year of activity of the research team.

A mixed-method study, that included a qualitative and a quantitative analysis, was set up to achieve an overall understanding of the topic. The qualitative study was based on semi-structured interviews and medical examinations conducted by a team of three physicians (a professor in occupational medicine with many years of clinical experience and two occupational medicine resident doctors), each of whom independently noted key work history data and recorded the women’s statements. Full notes were shared with the rest of the team on a case-by-case basis to enhance credibility, transferability, reliability, and determinability. To maintain interview confidentiality, medical histories were not recorded. In October 2021, a pilot study was undertaken to test the understanding of the open-ended questions and the effectiveness of the qualitative analysis. No modifications were required. All interviews lasted on average between 45 and 90 min. At the end of the medical examination, the doctors compared the results of their annotations, analyzed the qualitative data, and drafted a letter containing recommendations for the doctor in charge of the worker’s health surveillance in order to favor her RTW.

A quantitative, cross-sectional study was designed to compare the health status (work ability, fatigue, sleep status, anxiety, depression, and happiness) of women with BC with that of a control group sized 5n. Inclusion criteria for control women were: (i) female gender; (ii) age ± 1 year; (iii) no cancer history; (iv) currently working; (v) no night shifts. The questionnaires were tested during promotional activities carried out in the workplace by the Catholic University [23]. In the same year (2022), controls were invited to fill in the questionnaire during the regular medical examination in the workplace that Italian law makes mandatory for workers exposed to occupational risks. By means of a standardized questionnaire, BCSs were also asked to rate the level of organizational justice they perceived in the workplace before their illness. No control group was available for this assessment; consequently, the data were used for internal comparisons.

Sample size was not critical for qualitative analysis. For quantitative analysis, taking into account that, in our recent study, the prevalence of anxiety in active workers was 11.4% and that of depression was 14.1% [24], we calculated that a sample of 21 BCSs would allow us to see a significant difference for anxiety, while one of 25 BCSs would reveal a significant difference for depression (alpha 0.05, beta 0.2, power 0.8), if these symptoms were present in 50% of the sample. The calculation was carried out using Clincalc (https://clincalc.com/stats/samplesize.aspx, accessed on 20 June 2023). Since 32 women had asked to participate in the first year, the sample size was deemed to be sufficient.

### 2.2. Data Collection

Before the medical examination, the women underwent a semi-structured interview designed to reveal the factors that could hinder their RTW, those that could facilitate it, and the occupational environment conditions the worker had experienced before the illness. The factors facilitating and those hindering RTW are widely discussed in the literature and have recently been considered in a conceptual framework to which we have referred [22], with particular emphasis on factors related to adjusting strategies, in a clinical perspective [25]. Logically, the starting point is the analysis of occupational risk in the broadest sense, that is, both the assessment of environmental problems and organizational conditions in the workplace. This is followed by the assessment of the physical and psychological state of the BCS in relation to the therapies given and the coping strategies in relation to her motivational state. Thus, it is necessary to gather evidence on the type of intervention that can be suggested to implement the RTW and ensure its sustainability. For the interviews, we designed a draft questionnaire based on medical and occupational history. The semi-structured interview included topics related to disabilities, relationships with the management, and the quality of the work organization (Table 1).

All women who requested to be assisted in the RTW process in 2022 (n = 32) were interviewed. The outline-guide for the semi-structured interview is reported as a Appendix A, whilst details on the occupational history is in Appendix A.

Patients were also asked to complete the organizational justice questionnaire (described in the following section) with reference to their occupational status prior to the onset of the disease.

In the interview, to investigate the workers’ attitude towards work, which was often ambivalent due to intersecting favorable factors and obstacles, the patients were asked the question: “Do you think that going back to work can improve your condition?” The subjects who answered “yes” were classified as having an optimistic approach, while those who answered “no”, “probably no”, or “I don’t know” were reported as having a pessimistic approach.

### 2.3. The Questionnaire

At the end of the interview, the women were asked to fill in a questionnaire containing some standardized tools.

The questionnaire investigated different aspects of work.

Their current work ability compared with their highest work ability before contracting cancer was investigated with a single question, drawn from the Work Ability Index (WAI) [26]: “Supposing your best work ability has a value of 10 points, how many points would you give your current work ability? Scoring ranged from 0 = ‘I am currently unable to work’ to 10 = ‘My work ability is currently comparable with my work ability before cancer diagnosis’”.

Anxiety and depression were assessed using the Italian version [27] of the “Goldberg’s Anxiety and Depression Scale” (GADS) [28], with reference to the previous 10-day period. The GADS is composed of two scales of nine binary questions each; one point is awarded for each positive answer. A score of 5 or more on the anxiety subscale, or 2 or more on the depression subscale, indicates suspected clinically evident anxiety or depression [28]. Cronbach’s alpha in this study was 0.79 for the anxiety subscale and 0.82 for the depression subscale.

Sleep quality was studied using the Italian version [29] of the “Pittsburgh Sleep Quality Index” (PSQI) [30], with reference to the previous month’s habits. The scale consists of 18 items forming 7 components, each of which has a score ranging from 0 to 3. The minimum score is 0, and the maximum is 21. Higher scores indicate worse sleep quality. According to the original version of the scale [30], subjects with scores of 5 or more were defined as “bad sleepers”. Cronbach’s alpha of the questionnaire in this study was 0.86.

Fatigue was measured with the Fatigue Assessment Scale (FAS) [31]. The FAS consists of 10 questions; the unidimensionality of the questionnaire has been established [32]. Each response was graded on a 5-point Likert scale from 1 (“never”) to 5 (“always”). Scores on questions 4 and 10 were inversely recoded. By adding up the scores of all the answers, a total FAS score was obtained ranging from 10 to 50. A score ≥ 24 has been proposed as a cut-off for classifying fatigue on the FAS [33]. Cronbach’s alpha in this study was 0.85.

Happiness was measured using the Abdel-Khalek single item (“Do you feel happy on the whole?”) answered on an 11-point scale (0–10) [34]. Scores were dichotomized using the median value as cut-off.

Lastly, workers were given the scale concerning organizational justice at work prior to the onset of their current health problem. We used the Italian version [35] of Colquitt’s Organizational Justice Measure (OJM) [36]. This instrument consists of 20 questions, each graded from 1 to 5, from “very little” to “very much,” corresponding to 4 subscales of procedural (7 items), distributive (4 items), interpersonal (4 items), and informational justice (5 items). The overall final score ranges between 20 and 100. Cronbach’s alpha was 0.94 for the questionnaire (0.89 for procedural justice; 0.91 for distributive justice; 0.92 for interpersonal justice; and 0.87 for informative justice).

### 2.4. Qualitative Data Analysis

The first part of the medical interview focused on clinical aspects, physical health condition, and the outcome of treatment. Work history investigated relationships with colleagues and superiors and the perception of organizational justice. Lastly, the interview investigated knowledge of the benefits Italian law grants women with BC and the worker’s attitude towards returning to work. This multi-faceted interview gave us the opportunity to identify the factors that hindered or could facilitate the RTW.

COREQ criteria (consolidated criteria for reporting qualitative research) were followed in the design of the qualitative data [37]. Qualitative data were analyzed according to a modified version of the six-phase Thematic Analysis provided by Braun and Clarke [38]. Once the information from the medical interviews had been collected, medical histories and statements written by the physicians were compared and codes were developed inductively, revised, and transcribed in a well-organized database (phase 1). Peer debriefings were then performed based on an overall reading of each transcribed interview to generate an initial data codification (phase 2). The codes were later clustered based on similar and parallel findings and were grouped together to create broader themes. Repetitive analysis using a team approach helped increase the reliability of data, and the initial coding framework included negative and positive aspects of the RTW experience themes (phase 3). According to observations collected in the literature concerning the RTW of BC women [8,10,39,40,41,42,43,44,45,46,47] and textual fragments collected from interviews, a thematic map was drawn up and sub-themes related to the two main themes were inserted (phase 4). Afterwards, we classified themes as ‘barriers’ (negative aspects) and ‘facilitators’ (positive aspects) (phase 5). A final report was written containing the results of the study (phase 6). At the end of the analysis, the sentences were translated from Italian into English.

### 2.5. Statistical Analysis for the Case-Control Comparison

Descriptive statistics were applied to questionnaire scores; continuous variables were expressed as mean ± standard deviation (SD); and categorical variables were displayed as frequencies. All data were analyzed for normality of distribution using the Kolmogorov–Smirnov test of normality. Student’s T test or Mann–Whitney U test and χ^2^ test were applied to compare cases and controls. Odds ratios were calculated using logistic regression, setting the category (cases or controls) as the independent variable and the disorder as the dependent. Correlation between the variables of interest was studied with Spearman’s rho. Statistical analyses were performed using IBM SPSS Statistics for Windows, (Version 26.0. Armonk, NY, USA: IBM Corp., release 15.0).

### 2.6. Ethical Considerations

Each woman was administered an informed consent form concerning the study aims and the purpose of the interview. Complete anonymity and confidentiality were guaranteed for the opinions expressed. Participation in the study was voluntary and the women were given assurance that that they could withdraw from the interviews at any time without being obliged to give an explanation. They were free to pass on the letter addressed to the occupational doctor containing advice for their return to work, or to ignore the advice. The occupational medical examination was provided free of charge. The project was approved by the Research Ethics Committee of the Università Cattolica del Sacro Cuore of Rome (project nr. 4672, approved 14 February 2022).

## 3. Results

In 2022, 32 women asked to participate in the project. Their age (mean ± standard deviation) was 50.03 ± 8.99 years. The quantitative analysis control group, obtained from the population that annually underwent medical examination in the workplace by doctors of the Catholic University, consisted of 160 women who had agreed to fill in the questionnaires. Their age was 48.44 ± 8.91 years.

The results of the interview are reported in Table 2. Thematic Analysis allowed us to recognize themes concerning person-related, company-related, and society-related factors. Each of these themes was able to act as an obstacle or a facilitator for returning to work. In the course of the interviews, we systematically investigated which of these factors constituted a barrier, so that we could suggest the ergonomic measures or work reorganization that the doctor in charge of health surveillance could implement to remove these factors and facilitate RTW. Likewise, we suggested enhancing facilitating factors, so that the company physician could increase the woman’s well-being during the RTW process.

### 3.1. Barriers

The trauma associated with BC could be divided into three main categories:(i)Surgical (e.g., impairment and paresthesia in extremities, cosmetic damage, arm lymphedema, shoulder or arm pain syndromes, reduced strength, difficulty in lifting the arm):

“When my arm is immobile for a while, or when I wake up in the morning I often feel as if there are small needles in my shoulder and right arm. This also happens to me at work when I am sitting at my desk”;

“I have difficulty lifting or moving heavy objects, sometimes even when I am cleaning the house or have to put books on high shelves. Sometimes I drop things from my hand because my grip gives away”;

“My arm has swollen causing me some problems with movement, but I have learnt to manage it over time. The thing that bothers me most is that someone in the office might notice the difference between my two arms”.

(ii)Medical (e.g., iatrogenic menopause, loss of fertility, memory disturbances, speech errors):

“I get tired easily and have trouble concentrating, although I have already been back at work for a few weeks”.

(iii)Psychological (related to fear of the future resulting from the diagnosis of cancer which is rekindled with each new symptom; mental fatigue especially after relapses; difficulty in communicating needs):

“I’m tired and aware of the advanced stage of my disease, I’m severely depressed”.

Physical factors hindering RTW were frequently reported:

“I can’t move heavy objects, especially if I have to place them on high shelves”;

“My arm gets tired because I have to shift loads many times a day. Even if the loads don’t weigh more than 10 kg, I have to move them several times a day”;

“I can’t stand up for the whole work shift”.

Neuropsychological and cognitive factors, including lack of attention and memory, tiredness, and rapid exhaustion, were very common:

“I don’t want to go back to work because I’m tired and aware of the advanced stage of my disease, I’m severely depressed, and I don’t feel like working”;

“I fear that my work performance has decreased”;

“I often feel tired, and have a hard time concentrating”;

“I don’t want to hinder my colleagues in achieving company goals”

Personal aspects that were probably related to motivational blocks and psychosocial problems were able to hinder RTW:

“I think maybe I have been working too much, I already have 35 years of service, maybe I need to stop”;

“The work I was doing was too modest for my abilities, I wish the company would offer me something better now”;

“I don’t feel completely well, I need more time to fully recover”;

“The work environment was full of tension; I don’t feel like going back into a stressful environment”.

Environmental ergonomic factors were often reported:

“The trolleys I have to use at work have faulty wheels and I have even more trouble moving them”;

“I know that working nights could make my condition worse, but I have to continue working about two night shifts a week. I can’t refuse because I am hired as a freelancer”;

“I’m afraid I can’t handle the heavier workloads that I managed previously”;

“My incorrect posture at work will make my health condition worse”; “I don’t think I can stand night shifts any longer”;

“My work schedule and the need for in-person meetings may destabilize my work-life balance with regard to ongoing treatment”;

“Changing my workplace is a source of severe stress for me”;

“I have to travel a long way to get to work. Commuting is a challenging factor for RTW”.

Lastly, the fear of SARS-CoV-2 infection was also reported as a cause of concern:

“Being in the office with my colleagues makes me afraid of contracting COVID-19”.

Negative factors related to company staffing policy influenced two opposite but equally dysfunctional corporate behaviors in which the disease was either scotomized or emphasized. In the first case, the illness and disability associated with BC were ignored. As a consequence of this approach, a BCS should return to her place in the company after treatment, provide the same services as before, and continuously improve her productivity. Company production standards must not be jeopardized by the conditions of one of the workers.

“They say the problem is over. My place in the company is still the same, I must provide the same services as before and continuously improve productivity. But I just can’t do it”.

The opposite situation, in which management manifests intense emotional involvement, can also be damaging for a BCS who is returning to work since the person who has had a problem may be put aside because she has a disability, is fragile, or is considered unreliable.

“I was sidelined because of my situation. They considered me unreliable and not suitable for being involved in projects because I had no future in the company. They preferred to count on someone who could offer them greater reliability”;

“When they learned of my illness, they took away the job I was doing before. They effectively demoted me”.

In some cases, the disease led to a difficult family situation, or aggravated existing problems.

“My partner tried to take advantage of my condition to obtain custody of the children in the separation lawsuit”.

Our survey revealed a paradoxical effect when sometimes, among the obstacles to RTW, women reported the presence of laws that had been designed for their protection. Italian law provides several benefits for patients such as BCSs, who the National Institute of Social Security consider to be severely disabled. These patients can be transferred to the place of employment closest to their home; they have the right to be assigned to tasks appropriate and compatible with their reduced work capacity, without any reduction in salary; they can ask not to be assigned to night shifts; and they can request a switch from full-time to part-time, without losing their job. They can ask to telework from home during treatment. Moreover, many companies have developed forms of teleworking, whereby through employer–employee agreements, female workers with BC (as well as healthy ones) can take advantage of work organized into phases, cycles, and objectives. During the pandemic, female workers with immunodepression resulting from cancer treatment were allowed to enjoy telecommuting, even outside of corporate agreements. This wide range of benefits is aimed at encouraging RTW. However, in some cases, patients reported that the welfare measures had been used to create difficulties and cause them to quit their jobs.

“When the manager learned that I was sick, he arranged for my transfer to a work location closer to my home. I know I can ask to be moved closer to home because I am severely handicapped, but I have no intention of doing that. I don’t want to transfer; I don’t feel able to fit into a work environment that I don’t know”;

“When the pandemic broke out, the manager sent me to a medico-legal board to assess whether I could continue working in contact with children, and the board decided that I was unfit for work because of the risk of infection”.

Moreover, welfare benefits are not equally accessible to all women. In fact, some benefits apply only to civil servants, while others are extended to private employment, although they are limited to workers with a permanent contract. Workers with temporary contracts are not entitled to any benefit. This disparity of benefits between categories of female workers was perceived as discrimination:

“Access to smart working from home was granted only to people with oncological conditions undergoing treatment with immunosuppressive drugs, thus creating a human capital management disparity between companies that provide flexible work and others”;

“I had a fixed-term contract that expired while I was on sick leave, and it was not renewed”.

### 3.2. Facilitators

Physical facilitating factors related to treatment were frequently reported. Innovative surgical treatment involving immediate aesthetic reconstruction for women after breast removal was designed to shorten recovery time:

“Having my breast reconstructed immediately [after mastectomy] prevented damaging my female appearance and the image others have of me”.

Work engagement was a relevant aspect too:

“I have always had a great desire to work, and I want to find it again”.

The emotional and practical social support offered by employers and colleagues was also often reported. The support of colleagues and employers was of the utmost importance for BCSs. Moreover, organizational policies that included flexibility in schedules, the possibility of part-time work, and a gradual return to the workplace in order to avoid overload were also considered to be facilitators:

“My employer is extremely helpful in promoting a gradual return to work so that I can avoid excessive fatigue.”

Also, non-work relationships, such as family support, were considered to be an important mainstay of an RTW:

“My partner, who is also a nurse, helped me a lot”.

Participants reported relevant welfare-related factors facilitating RTW. Telecommuting was a useful organizational strategy that gave them the possibility of continuing to work even immediately after surgical treatment:

“Teleworking allowed me not to lose job opportunities because I continued to write articles and maintain correspondence from home”;

“The occupational physician in the workplace exempted me from the heaviest work when he learned about my condition”.

The themes that emerged from the qualitative analysis are listed in a Supplement to this article (Appendix A).

### 3.3. Organizational Justice and RTW

For the women participating in our study, the mean level of organizational justice perceived in their previous work environment was 63.6 ± 19.1. This score corresponds to 55% of the theoretically achievable maximum and is close to the value that can be measured in the workplace (e.g., among the various occupational categories in a hospital) [48].

We aimed to investigate whether workers’ expectations of RTW were associated with the level of organizational justice they had experienced in the past.

Seventeen women were convinced that RTW would improve their condition. Contrary to this optimistic attitude, fifteen women expressed a negative opinion or had doubts as to whether the RTW would improve their condition. Women with an optimistic approach to RTW perceived greater organizational justice in their work environment than women with a pessimistic approach (70.6 ± 18.1 vs. 54.4 ± 16.3, *p* < 0.05).

### 3.4. Quantitative Analysis

When asked to rate their current work ability compared with their ability prior to illness, the great majority of women stated that it had decreased significantly, thus resulting in an estimated mean 50% loss in work ability. On average, cases had significantly higher scores for anxiety, depression, sleep problems, and fatigue, and a lower happiness score than controls (Table 3).

We used Spearman’s rho to study the correlation between the quantitative variables (Table 4). Work ability was negatively associated with fatigue. The perception of organizational justice was negatively associated with anxiety, depression, and poor sleep quality and positively associated with happiness. Poor sleep quality was positively correlated with fatigue, anxiety, and depression and inversely correlated with happiness. Fatigue was associated with anxiety and depression and inversely with happiness. Anxiety and depression were correlated with each other. Depression was inversely associated with happiness.

Over 80% of BCSs had a low level of sleep quality, and 74.2% reported excessive fatigue. A total of 65% of BCSs were anxious and over 84% of them were depressed, compared with 15% and 28% of the controls, respectively. Also, the prevalence of low-level happiness was higher in BCSs than in controls (Table 5).

## 4. Discussion

This study obtained two main results. The first, and on the basis of the literature, probably the least expected, was that the main categories of factors influencing RTW (person-, corporate- and society-related factors) can play both facilitating and hindering roles, depending on specific workplace conditions. This variability requires a careful study of each individual situation and determination to provide personalized support in the recovery process. This project was specifically conceived to help the doctor in charge of occupational health surveillance and his/her company to achieve greater effectiveness in the worker’s recovery process.

The second result was the measurement, using standardized methods, of the difference in working capacity and mental health (sleep, fatigue, anxiety, depression, happiness) between female workers returning after illness and working women. This result could be useful for the company doctor when grading the workload according to the BCS’s residual abilities and assigning occupational tasks that she could cope with. In the sample of BCSs, the women reported that their working capacity was reduced by 50% compared with the level before the disease. This could give rise to the phenomenon known as presenteeism, which must be properly managed. A loss in productivity in BCSs is a well-known factor. Female workers affected by BC lose, on average, one-fifth of earning capacity [49]. In a French retrospective study, one-third of BC patients continued working during treatment, and 89% returned to work. Three-quarters of the BCSs reported a decline in work capacity one year after RTW and one out of five manifested a persistent decline in work capacity two years after the diagnosis [50]. An estimated 8% productivity loss due to presenteeism has been observed in a BCS group [51].

A number of qualitative studies on this topic have already been published. A meta-synthesis of qualitative studies identified three major themes, roughly corresponding to the ones we observed: personal factors, employment factors, and wider contextual factors including family, social, and cultural variables [52]. Thematic Analysis enabled us to identify factors related to the individual, the company, and society. The experiences of female workers demonstrated that each of these classes of factors could favor or hinder the RTW. Another systematic study reached similar conclusions to the ones reported in this study. Disease-related factors such as poor health and fatigue, work-related factors such as hard physical labor, and psychological factors such as depression and emotional distress were listed among impediments to RTW. On the contrary, workplace support, social support, and family support were the main factors that helped BCSs return to and carry on with their jobs [12]. Another meta-synthesis adopted a different point of view by trying to interpret the personal evaluations of women. The study identified four major themes: lack of meaning after the cancer diagnosis; concerns and considerations before RTW; motivations for returning to work; and working life after cancer [53]. Data from a French cohort study confirmed that many of the BCSs reordered their life priorities after the diagnosis [54]. In our study, which was more oriented toward workplace operating conditions, women’s motivations and concerns played an important role, which we tried to summarize in opposing pessimistic or optimistic attitudes to RTW. Our study design did not enable us to assess the participants’ life after resuming work. However, we have already scheduled a one-year follow-up study to evaluate the quality of life of women who return to work.

Medical examinations made it possible to objectify residual physical problems after treatment. Previous studies had already included physical limitations among the obstacles to RTW [43,44,46,55]. Pain—especially in the axilla and arm on the affected side—plays an important role in the quality of a patient’s life [41] and in RTW [56]. Pain can be a direct consequence of the disease or arise after surgery, radiotherapy, or chemotherapy [57]. In recent years, however, advances in medical technology and more specific therapies have made these side effects less frequent [58]. Moreover, personalized rehabilitation interventions using a variety of techniques (e.g., exercises [59], lymphatic drainage [60,61]) can significantly reduce patient discomfort and increase fitness for work. To date, there is a lack of evidence of the effectiveness of rehabilitation interventions supporting the RTW of BCSs [62]. Studies have also pointed out that RTW can be supported by a series of ergonomic measures [63]. Carrying out these interventions in the workplace obviously requires close collaboration and communication between the occupational physician responsible for health surveillance, the company prevention and protection service, the company management, and specialists capable of carrying out the necessary environmental or individual measures. Since effective collaboration is essential, all the examinations we conducted led to a report contained in a consultation letter for the occupational physician of the company the woman worked for. Our specific aim was to improve this type of communication, so that each individual patient received the crucial intervention required.

We also measured the quantitative difference in health status between BCSs returning to work and the controls. The loss of work capacity reported by our sample was significantly related to fatigue. However, BCSs also had significantly higher levels of sleep problems, anxiety, depression, and unhappiness compared with healthy peers.

More than 80% of the women who asked us to help them return to work had poor sleep levels. Sleep quality assessed at the time of the medical examination was directly associated with anxiety, depression, unhappiness, and fatigue. It also seemed to depend, inversely, on the level of organizational justice experienced at work prior to illness. Previous studies have reported that sleep problems, including insomnia (difficulty falling asleep, nocturnal awakenings, and non-restorative sleep), are commonly experienced by patients with cancer [64], especially BC [65], and continue to impact the quality of life even several years after treatment [42]. One study suggested that over 60% of patients with BC experienced reduced sleep times and frequent sleep disturbances [66]. A longitudinal cohort study revealed that about 80% of BCSs experienced elevated insomnia symptoms with no significant improvement at more than a year after diagnosis [67]. Several etiologic factors may play a role in the development of sleep problems, including intrinsic stress and uncertainty, the side effects of treatment (e.g., endocrine therapy [68], breast surgery [69] chemotherapy and/or radiotherapy [70]), and the psychological impact of BC diagnosis (e.g., depression, anxiety, fear of recurrence [71]). Sleep disruption may also remain several years after BC treatment in the follow-up period [72,73], probably due to the molecular mechanisms of the host–tumor interaction [74]. Moreover, there appears to be a mutual relationship between cancer and sleep disruption, since cancer seems to promote disrupted sleep and poor sleep promotes tumorigenesis and cancer progression [74]. Sleep disturbance among cancer patients is a potential contributor to reduced work performance and higher absenteeism [75]. Sleep is a critical factor for the health and safety of workers and for their well-being [76]. For this reason, we studied the BCSs’ sleep and informed the company occupational doctors of the workers who needed assistance in getting back to normal sleep. The association of sleep problems with mental health issues should lead to combining stress counseling with proper sleep hygiene. It is also worth noting that poor sleep quality was linearly associated with a low level of perceived justice in the work environment. Consequently, the employer’s first duty to BCSs is to improve work organization.

Over 74% of the participants in our study were fatigued. Fatigue was significantly correlated with a loss in work ability. Cancer-related fatigue (CRF) is the commonest and most disturbing symptom for BC survivors [77] and often prevents them from returning to work [78]. It was reported as a persistent problem for 24% of a group of BCSs during a 2-year follow-up period [79]. CRF is commonly referred to in terms of changes in weight, menopausal symptoms, coping, social support, and biochemical changes in BC survivors [80]. Fatigue has been shown to be a consequence of active treatment, but it may also persist into post-treatment periods [81]. Growing evidence suggests an inflammatory basis for CRF, which is closely linked to alterations in the neuroendocrine and immune systems [82]. However, a correlation with other factors has also been observed. Depression and pain emerged as the strongest predictors of fatigue, which was also influenced by sleep disorders [83]. Another study showed that sleep disturbances, emotional symptoms, and neuromuscular fatigability were the most important CRF predictors in cancer patients [84]. This evidence highlights a complementary aspect of our research by confirming a strong bidirectional relationship between sleep problems and CRF. On returning to work, excessively fatigued BCSs need to be helped to recover not only by reducing the physical or mental load associated with their work, but also by improving their sleep. Such a measure also has a purely economic justification, since it would improve the BCS’s ability to work.

Our sample of BCSs had significantly higher scores for anxiety and depression than healthy controls. Anxiety and depression were associated with fatigue and sleep problems. On analyzing the cut-off values, two out of three subjects appeared to be anxious and four out of five appeared depressed. When assessing these percentages, it should be remembered that the GADS questionnaire is a screening tool: a worker who passes the cut-off has a 50 percent chance of developing the condition and the risk of being diagnosed as anxious or depressed increases rapidly as the score increases. In the literature, approximately 11–16% of BCS patients experience combined symptoms of anxiety and depression, and cancer-related anxiety is associated with the risk of sleep disorders [85]. Mental health disorders are a strong obstacle to RTW. Anxiety was a factor significantly associated with a non-return to work in the BCSs of the French Seintenelles study [50]. Depressive symptoms were associated with impaired RTW after one year in a German study [14], and also in a study conducted by a Chinese researcher [86]. In another French study, BCSs treated with antidepressant/anxiolytic drugs returned to work after a longer period than other patients [87]. Depression is also associated with higher productivity loss after RTW: an Australian study showed that productivity loss was approximately fourfold higher in the depressed group than in the non-depressed group [88]. Conversely, RTW was reported to be associated with improved mental health in cancer survivors, when compared with nonworking patients who, on the contrary, had higher levels of depression, anxiety, and distress [47].

As expected, in our sample, women with cancer had a lower average level of happiness in life than healthy women. The traumatic experience of cancer is known to be associated with sleep deprivation [89]. However, according to previous studies, positive affective states (e.g., joy, happiness, vigor, positive mood) in BCSs may be the result of a change in life perception following the discovery of simpler and more important aspects of everyday life [90]. In our sample, happiness was associated with fatigue at work, sleep problems, and depression; moreover, it was greater in women who came from a work environment that they perceived to be organized in a fair and just way. The inverse correlation between fatigue and happiness is worthy of attention. BC is associated with weight gain, muscle atrophy, and weakness. A reduction in physical activity can increase the side-effects of treatment, fatigue, and psychological factors such as depression, anxiety, body-image, and unhappiness [91]. Conversely, higher physical activity and daily step counts are associated with lower depression levels and stronger perceptions of happiness [92].

Measures to enhance RTW for cancer patients can be classified according to physical, psychological, vocational, and multidisciplinary interventions; the latter (which included physical, psycho-educational, and vocational components) led to higher RTW rates than routine care [93,94]. Conversely, interventions based only on physical rehabilitation, were less satisfactory. In the past, there have been few rehabilitation programs to help BCSs keep or return to their job. In the few projects published to date, a recent meta-analysis failed to find conclusive evidence of improvement in work outcomes [95]. The aim of our vocational intervention was to provide the occupational doctor with the information needed to ensure that the worker is given the personalized support required. To the best of our knowledge, no such projects are currently available in the literature.

The main strength of our study lies in our method, which involved occupational physicians both in the collection of the qualitative data and the formulation of advice. In the workplace, another occupational physician is responsible for correcting the environmental conditions and work organization, as well as helping the worker to take advantage of welfare measures and follow a personalized path that should lead to occupational recovery.

Another strength was our productivist approach, which is a characteristic of occupational health. By conducting an in-depth qualitative analysis of the factors that could hinder or favor RTW in each individual case, our aim was to indicate to the company doctor the most effective interventions for favoring a return to work, whether they be physical, psychological, vocational, or mixed. Using the same productivist point of view, we evaluated the difference between the women who were recovering from BC and the healthy female workers. The BCS recovery process entails an economic cost that must also be carefully evaluated in order to guarantee that the intervention can be sustained. A possible development of this study is to analyze the impact of the work capacity deficit in terms of reduced productivity, one year after the RTW. Another possible development is to assess which, among the many factors studied, are significantly associated with sustained RTW after time.

The limitations of this study include sample size, which was nevertheless larger than that of other qualitative studies [39,43,96,97,98] that were sometimes limited to just six or seven cases [46,97]. Apparently, ours is the largest qualitative study in the literature; what is more, none of the qualitative studies published to date were combined with quantitative analysis, and none with personalized job adaptation proposals. The mixed method and analysis of past work situations, current physical and mental health status, and individualized adjustments that would have been helpful in resuming work represent the strengths of the study.

Another weakness was the selection of patients from a single research center, which might restrict the validity of our results in other situations. It should be noted, however, that the health organizations involved are among the largest in our country.

Qualitative studies are often criticized because they fail to use standardized methods and the results may depend heavily on the opinions of the researchers. For this reason, the qualitative survey was accompanied by a quantitative one, for which a control group of 5n size was selected from all women undergoing occupational health screening in the workplace. The quantitative analysis showed that women returning to work have significant deficits and must be supported to gradually recover their productive capacity.

We intend to develop this study by monitoring return to work one year after the first medical examination and by continuing to observe the women who ask for assistance. Responses from company occupational doctors will help us to make our indications more precise and useful. We are convinced that the communication of proposals to occupational doctors must gradually become a good practice for all patients who are about to re-enter the workplace after being discharged from hospital.

## 5. Conclusions

Since RTW after BC is associated with increased survival rates and a better quality of life, it must be encouraged. Although both society and companies share this goal, our study nevertheless showed that there is much that can be improved in the RTW process since some benefits may be applied in an inappropriate and counterproductive manner. There is no exact formula for achieving effective RTW: all occupational medicine intervention must be ‘tailor-made’ for each patient through careful selection and implementation. Our study showed that, upon the resumption of work, women have an average loss of work capacity of 50% compared with pre-illness and increased ORs of sleep problems, fatigue, anxiety, depression, and unhappiness compared with healthy women of the same age. The process of reintegration into work is long and complex and prevention operators must operate in the best possible way to make it complete and sustainable. Hospitals, which have greatly improved diagnostic procedures and treatments and now achieve a physical and psychological recovery that was unthinkable in the past, should extend their tasks to indicating which of the many possible paths corporate occupational physicians should follow for an effective return to work after BC. Companies, as part of their social responsibility, will be happy to address changes in work organization if they are aware that their efforts will help to achieve the quickest and most complete recovery of the occupational skills and experience of a worker who has undergone the stressful experience of breast cancer.

## Figures and Tables

**Table 1 healthcare-11-02343-t001:** Items of the semi-structured interview.

Objective Investigated	Items
Health problems	“Are you suffering from any disabling physical problems?”
	“Are you suffering from any disabling mental problems?”
Relationship with the management	“What prospect did your superior propose for your RTW?”
Quality of Work Organization	“How do you perceive the organizational justice of your workplace?”
	“Please, fill in the Organizational Justice Questionnaire”

Notes: RTW: Return to work.

**Table 2 healthcare-11-02343-t002:** Barriers and facilitators for RTW from qualitative data.

	Themes	
	Barriers	Facilitators
Person-related factors	Physical problems (pain, fatigue) Motivational blocks Cognitive and neuropsychological problems (reduced concentration, decreased performance, apathy)	Surgical breast reconstruction Work engagement
Company-related factors	Work overload Work underload Environmental and ergonomic factors Inadequate shifts Employer’s request for work ability assessment	Policies for RTW Ergonomic and schedule adjustments Social support from colleagues and superiors
Society-related factors	Unequal access to welfare benefits Family conflict	Legal and welfare benefits for workers with cancer Telecommuting, teleworking Social support from family members

Notes: RTW, Return-to-work.

**Table 3 healthcare-11-02343-t003:** Comparison between cases and controls.

Variable (Score Range)	Cases		Controls		*p* Value
	Mean ± s.d.	Median (IQR)	Mean ± s.d.	Median (IQR)	
Work ability	5.00 ± 2.44	5.00 (3.25, 6.00)			
Fatigue (10–50)	29.03 ± 8.69	31.0 (21.0, 36.0)	18.46 ± 4.91	18.0 (15.0, 21.5)	<0.001
PSQI (0–21)	9.0 ± 4.0	9.0 (7.0, 12.0)	6.0 ± 3.2	6.0 (3.0, 8.0)	<0.001
Anxiety (0–9)	5.66 ± 2.63	6.0 (4.0, 8.0)	1.84 ± 2.29	1.0 (0.0, 3.0)	<0.001
Depression (0–9)	4.25 ± 2.70	4.5 (2.0, 6.0)	1.27 ± 1.81	0.0 (0.0, 2.0)	<0.001
Happiness (0–10)	5.94 ± 2.56	6.0 (4.5, 8.0)	7.44 ±1.75	8.0 (7.0, 9.0)	0.003

Notes: s.d.: standard deviation. IQR: InterQuartile Range.

**Table 4 healthcare-11-02343-t004:** Correlation between quantitative variables (Spearman’s rho).

Variable	Work Ability	Justice	PSQI	Fatigue	Anxiety	Depression	Happiness
Work ability	1						
Justice	0.148	1					
PSQI	−0.200	−0.440 *	1				
Fatigue	−0.447 *	−0.320	0.528 **	1			
Anxiety	−0.002	−0.539 **	0.720 **	0.444 *	1		
Depression	−0.183	−0.629 **	0.619 **	0.594 **	0.603 **	1	
Happiness	0.115	0.416 *	−0.449 *	−0.426 *	−0.334	−0.535 **	1

*. Correlation is significant at the 0.05 level (2-tailed). **. Correlation is significant at the 0.01 level (2-tailed).

**Table 5 healthcare-11-02343-t005:** Comparison of characteristics between cases and controls.

Characteristic (n)	Cases vs. Controls (n, %)	ODDS RATIO (CI95%)	Pearson χ^2^	*p* Value
Bad sleeper (190)	27 (84.4%) vs. 111 (70.3%)	2.29 (0.83; 6.30)	2.67	0.102
Fatigued (187)	23 (74.2%) vs. 39 (25.0%)	8.63 (3.57; 20.84)	28.24	<0.001
Anxious (185)	21 (65.6%) vs. 23 (15.0%)	10.79 (4.60; 25.34)	37.37	<0.001
Depressed (188)	27 (84.4%) vs. 44 (28.2%)	13.75 (4.98; 37.97)	35.65	<0.001
Unhappy (185)	23 (71.9%) vs. 73 (47.7%)	2.80 (1.22; 6.44)	6.19	0.015

## Data Availability

The data sets generated and analyzed during the current study are not publicly available due to the sensitive and identifiable nature of our qualitative data but are available from the corresponding author on reasonable request.

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
