# Peer review of "Supporting Return to Work after Breast Cancer: A Mixed Method Study"

_healthcare, 2023, doi:10.3390/healthcare11162343_

Round 1
Reviewer 1 Report
Dear author's
I was pleased to review your manuscript and I have the following comment's<
- Please explain the novelty of your study.
- The section Introduction should introduce the reader with the general concept of the article, so please focus on BC and return to work.
- "Breast cancer is the most commonly diagnosed cancer in women and affects many 611 working-age females" - This conclusion is not a result of your research. Please remove this sentence form conclusion.
- The sample is relatively small, it is very difficult to draw solid conclusion.
- Please highlight the limitation of the study.
- Minor punctuation edits.
Author Response
We thank the reviewer. The point-by-point responses are in the attached file.

Reviewer 2 Report
First, I would like to thank the editor for the opportunity to review this manuscript. The work appears to be correct, although it needs improvement in order to contribute to the scientific literature. I would ask the authors to revise the qualitative design.
Introduction
· In general terms, the introduction is correct, although I consider that it has few bibliographical references to justify the study. The ideas are dispersed by not finding a clear focus, at least that is my feeling. I would recommend the authors to revise the introduction, incorporate at least 5 more references and make clear the purpose of this study before ending the section.
· Exactly what does it mean to describe factors that make it difficult or easy to return to work? And analyzing health status? Both seem very generic statements. (Lines 78-82)
Materials and Methods
· Why is the semi-structured interview constructed this way and not in another way?
· Please use only two decimal places to mark Cronbach's alpha.
Results
· Why use Pearson and Spearman and not one of the two?
· Very little importance is given to quantitative results and they do not really contribute much information to the study...
Discussion and Conclusions
· I don't have much to contribute. These sections seem to me to be correct, although I must say that they speak of results in a very forceful way when my perception is that they are inferred too much from the qualitative analysis and from a semi-structured interview that does not explain well why it has been designed in this way....
Moderate editing of English language required
Author Response
We thank the reviewer. The point-by-point response are in the attached file.

Round 2
Reviewer 1 Report
Dear author's
Thank you for your response. It still unclear the data collection that you performed in this study.
Interview with no? patients
Questionnaire for auther ..
Please make this section more clear.
Author Response
Reviewer#1, round #2
Dear author's
Thank you for your response. It still unclear the data collection that you performed in this study.
Interview with no? patients
Questionnaire for auther ..
Please make this section more clear.
R: The number of patients who requested assistance was 32, as reported in the results section. We have not refused any patients.
We have attached Supplementary File 1 with the outline-guide for the structured interview, and Supplementary File 2 with details on the occupational history in Italian.
The manuscript now is as it follows:
“All women who requested to be assisted in the RTW process in 2022 (n=32) were interviewed. The outline-guide for the semi-structured interview is reported as a supplement (S1), whilst details on the occupational history is in S2."
